# SMURF1-Induced Ubiquitination of FTH1 Disrupts Iron Homeostasis and Suppresses Myogenesis

**DOI:** 10.3390/ijms26031390

**Published:** 2025-02-06

**Authors:** Xia Xiong, Wen Li, Chunlin Yu, Mohan Qiu, Zengrong Zhang, Chenming Hu, Shiliang Zhu, Li Yang, Han Pen, Xiaoyan Song, Jialei Chen, Bo Xia, Shunshun Han, Chaowu Yang

**Affiliations:** 1Animal Breeding and Genetics Key Laboratory of Sichuan Province, Sichuan Animal Science Academy, Chengdu 610066, China; xiongxia20120904@163.com (X.X.); lwss072@126.com (W.L.); yuchunlin1984@sina.com (C.Y.); mohan.qiu@163.com (M.Q.); zhangzengrong2004@163.com (Z.Z.); huchenming@126.com (C.H.); zhushiliang1994@163.com (S.Z.); yangli_sasa@163.com (L.Y.); penghan0706@163.com (H.P.); babalasxy@163.com (X.S.); qiaoqiaowo110@163.com (J.C.); allanbobo777@163.com (B.X.); 2Key Laboratory of Livestock and Poultry Multi-Omics, Ministry of Agriculture and Rural Affairs, College of Animal Science and Technology, Sichuan Agricultural University, Chengdu 611130, China; hanshunshun@sicau.edu.cn

**Keywords:** ferroptosis, FTH1, SMURF1, ubiquitination, skeletal muscle, iron homeostasis

## Abstract

Ferritin heavy chain 1 (FTH1) is pivotal in the storage, release, and utilization of iron, plays a crucial role in the ferroptosis pathway, and exerts significant impacts on various diseases. Iron influences skeletal muscle development and health by promoting cell growth, ensuring energy metabolism and ATP synthesis, maintaining oxygen supply, and facilitating protein synthesis. However, the precise molecular mechanisms underlying iron’s regulation of skeletal muscle growth and development remain elusive. In this study, we demonstrated that the conditional knockout (cKO) of FTH1 in skeletal muscle results in muscle atrophy and impaired exercise endurance. In vitro studies using FTH1 cKO myoblasts revealed notable decreases in GSH concentrations, elevated levels of lipid peroxidation, and the substantial accumulation of Fe^2+^, collectively implying the induction of ferroptosis. Mechanistically, E3 ubiquitin-protein ligase SMURF1 (SMURF1) acts as an E3 ubiquitin ligase for FTH1, thereby facilitating the ubiquitination and subsequent degradation of FTH1. Consequently, this activation of the ferroptosis pathway by SMURF1 impedes myoblast differentiation into myotubes. This study identifies FTH1 as a novel regulator of muscle cell differentiation and skeletal muscle development, implicating its potential significance in maintaining skeletal muscle health through the regulation of iron homeostasis.

## 1. Introduction

Skeletal muscle, composed of multinucleated myofibers originating from myoblast fusion, undergoes a process known as myogenesis to form functional muscle tissue [1]. This intricate process is tightly regulated by a network of transcription factors, with myogenic factor 5 (Myf5) and myoblast determination protein 1 (MyoD) playing pivotal roles in specifying the myogenic fate of progenitor cells [2,3]. Concurrently, myogenin (MyoG) and muscle regulatory factor (Mrf4) act in concert to drive the terminal differentiation of myoblasts and their subsequent fusion into myotubes, which mature into myofibers [4,5]. The importance of skeletal muscle extends beyond its role in initiating movement and supporting respiration; it is also essential for maintaining metabolic homeostasis [6]. Declines in skeletal muscle mass and function are associated with aging and various disease states, including cancer and diabetes [7,8]. Given its economic significance in meat-producing animals, skeletal muscle development and physiology have been the focus of extensive research. The identification of novel regulatory factors and mechanisms involved in these processes is crucial for advancing our understanding of muscle biology and holds promise for improving human health and animal productivity [9,10].

*FTH1* gene constitutes an essential subunit of ferritin, endowed with ferroxidase activity that catalyzes iron oxidation and facilitates cellular iron sequestration, thereby playing a pivotal role in maintaining iron homeostasis and protecting cells against oxidative injury [11,12]. In the context of ferroptosis, a recently characterized mode of cell death, FTH1 exerts a modulatory effect on the accumulation of lipid peroxides through its ferroxidase function, ultimately influencing the initiation of ferroptosis [13]. This regulatory role of *FTH1* in ferroptosis is particularly significant given its implications in various diseases [14]. Recent studies have demonstrated that *FTH1* plays a crucial role in cancer drug resistance [15]. In esophageal squamous cell carcinoma, *FTH1* has been shown to mediate paclitaxel resistance by inhibiting ferroptosis, thereby promoting cell survival [16]. Similarly, in the context of tumor immunotherapy resistance, *FTH1* contributes to the development of an inhibitory tumor immune microenvironment, which facilitates resistance to anti-PD-1/L1 immunotherapy by regulating immune cell infiltration and activation [17]. These findings indicate that *FTH1* holds a significant role in the physiological and pathological modulation of various diseases. 

Iron is vital for skeletal muscle function, as it participates in oxygen transport via myoglobin and hemoglobin, and plays a crucial role in ATP production through its involvement in the mitochondrial electron transport chain. Disruptions in iron metabolism can significantly impact muscle performance [18]. For example, studies have shown that iron deficiency leads to reduced myoglobin levels and impaired oxygen transport, resulting in muscle fatigue and reduced exercise capacity [19]. In recent years, iron-dependent programmed cell death has been found to play an important role in various diseases [20]. Ferroptosis, distinguished by its dependence on iron and reactive oxygen species (ROS)-induced cell death via lipid peroxidation, occupies a central position in skeletal muscle physiology and pathophysiology [21]. It profoundly affects muscle cell function and has been implicated in a spectrum of skeletal muscle disorders. Notably, within the aging skeletal muscle milieu, iron accumulation coupled with the induction of ferroptosis may hinder muscle regeneration and repair by downregulating the expression of satellite cell markers [22,23]. FTH1 plays a pivotal role in ferroptosis and ferritinophagy. In ferroptosis, FTH1 inhibits cell death by maintaining intracellular iron homeostasis [24]. When FTH1 expression is downregulated, it may promote the occurrence of ferroptosis. In ferritinophagy, FTH1 is a crucial component involved in the autophagic degradation of ferritin, which selectively degrades ferritin to release iron [25]. FTH1 interacts with NCOA4 to transport ferritin to autophagosomes for degradation. This process is essential for regulating intracellular iron levels and preventing cell damage induced by iron overload [26]. Given the central role of *FTH1* in the ferroptosis pathway and its involvement in skeletal muscle development and regeneration, it is conceivable that the modulation of FTH1 activity could provide a therapeutic strategy for skeletal muscle diseases. *FTH1*, a pivotal constituent of ferritin, stands in intimate association with iron metabolism [27]. An in-depth exploration into its molecular regulatory mechanism is instrumental in elucidating the precise functions of FTH1 in iron storage, release, and the maintenance of iron homeostasis. Alterations in FTH1 expression are intricately linked to the onset and progression of numerous diseases [14]. By delving into its molecular regulatory framework, we can unveil the role that *FTH1* plays in the disease process, thereby offering novel insights for the diagnosis and treatment of these conditions. This line of research paves the way for the discovery of new drug targets and the development of therapeutic agents tailored to combat related diseases. Furthermore, it provides a theoretical foundation for the advancement of innovative therapies, including gene therapy and cell therapy, thereby contributing to the expanding frontier of medical science.

Indeed, further elucidation of the precise mechanisms underlying FTH1′s involvement in skeletal muscle ferroptosis not only enhances our understanding of muscle physiology and pathophysiology but also offers promising avenues for the development of novel therapeutic and regenerative strategies in the management of skeletal muscle diseases, as well as other conditions where ferroptosis plays a critical role.

## 2. Results

### 2.1. FTH1 Knockout Mice Result in Muscle Atrophy and Weakness

To investigate the function of *FTH1* in skeletal muscle development, we generated mice with the conditional knockout of FTH1 specifically in skeletal muscle by crossing FTH1-floxed (fl/fl) mice with Myf5-Cre mice (Appendix A). Compared to control mice, morphometric assessments indicate a significant decrease in the weight of the gastrocnemius muscle (GAS), soleus muscle (SOL), extensor digitorum longus (EDL), and tibialis anterior (TA) muscles in the *FTH1* skeletal muscle-conditional knockout (cKO) mice (Figure 1A). Furthermore, the cross-sectional area of muscle fibers in the GAS is diminished in *FTH1* cKO mice, as illustrated in Figure 1B,C. Analysis of the mean fiber diameter demonstrates a higher proportion of thinner fibers in *FTH1* cKO mice (Figure 1D). Additionally, both the absolute force and specific tetanic force are decreased in these mice (Figure 1E), and their muscle performance during forced treadmill running tests is also impaired (Figure 1F).

### 2.2. FTH1 Knockout Induces Activation of the Ferroptosis Pathway in Myoblasts

To delve deeper into the molecular regulatory mechanisms governed by FTH1, we isolated primary myoblasts from *FTH1*-knockout mice. Subsequent RNA-seq analysis revealed that these *FTH1*-deficient cells exhibited significant enrichment in pathways related to muscle system processes, muscle cell development, and the regulation of iron ions (Figure 2A). KEGG enrichment analysis further identified a significant enrichment of the ferroptosis signaling pathway following *FTH1* knockout in the myoblast cells (Figure 2B). Western blot analysis demonstrated that the protein expression levels for the ferroptosis marker genes long-chain-fatty-acid–CoA ligase 4 (ACSL4) and cytochrome c oxidase subunit 2 (COX2) were significantly elevated in myoblasts with the *FTH1* knockout (Figure 2C). The qPCR analysis also found that ACSL4 and COX2 mRNA expressions were significantly increased in *FTH1* knockout myoblasts (Figure 2D). Consistent with the key hallmarks of ferroptosis—GSH depletion, lipid peroxidation, and Fe^2+^ accumulation—our findings revealed a significant reduction in GSH content along with marked elevations in the levels of MDA, GSSG, and Fe^2+^ in *FTH1* cKO myoblasts (Figure 2E–H).

### 2.3. FTH1 Modulates Myoblast Differentiation into Myotubes via the Ferroptosis Pathway

To further investigate whether FTH1 influences skeletal muscle development via the ferroptosis pathway, we assessed its effects on MyoG and MyoD expression. The results revealed a significant reduction in both mRNA and protein expression levels of MyoD and MyoG in FTH1-knockout myoblasts (Figure 3A,B). Immunofluorescence analysis of the MyHC expression of MyHC revealed that the FTH1 knockout significantly inhibited the differentiation of myoblasts into myotubes (Figure 3C). Subsequently, we administered the ferroptosis inhibitor deferoxamine (DFO) to FTH1-knockout myoblasts and observed that DFO treatment was capable of rescuing the ferroptosis induced by FTH1 deficiency (Figure 3D–G). The qPCR analysis demonstrated that DFO treatment markedly increased the downregulation of MyoG and MyoD expression induced by FTH1 knockout (Figure 3H). MyHC immunofluorescence showed that DFO treatment promoted a decrease in the myotube fusion rate caused by the FTH1 knockout (Figure 3I).

### 2.4. SMURF1 Directly Recognizes FTH1

Next, we investigated how FTH1 protein expression is regulated by those E3 ubiquitin ligases in UbiBrowser database. The results indicate that SMURF1 is identified as a potential E3 ubiquitin ligase for FTH1, exhibiting the highest predictive score among all candidate E3 ubiquitin ligases evaluated (Figure 4A,B). Subsequently, immunofluorescence analysis revealed that SMURF1 and FTH1 proteins share an identical subcellular localization in mouse myoblasts (Figure 4C). This observation was further substantiated by co-immunoprecipitation (Co-IP) experiments, which detected an interaction between endogenous FTH1 and SMURF1 in myoblast cells, but Atrogin1 (another protein reported to bind SMURF1) or control IgG did not precipitate, suggesting a functional association (Figure 4D). Additionally, Co-IP tests using epitope-tagged proteins in HEK 293T cells confirmed the specific interaction between Flag-labeled FTH1 and GFP-labeled SMURF1 (Figure 4E), reinforcing our initial findings.

### 2.5. SMURF1 Ubiquitinates and Degrades FTH1

To evaluate the involvement of SMURF1 in modulating FTH1 protein levels, we performed overexpression studies in HEK 293T cells. Wild-type (wt) Nedd4 overexpression led to a notable reduction in endogenous FTH1 protein levels (Figure 5A), hinting at a regulatory relationship. Conversely, expression of the catalytically inactive SMURF1C276S-Flag mutant did not impact FTH1 levels (Figure 5B), emphasizing the necessity of SMURF1′s E3 ubiquitin ligase activity for FTH1 protein destabilization. To further explore whether endogenous SMURF1 participates in erastin-induced FTH1 protein degradation, we transfected HEK 293T cells with shRNA targeting SMURF1. SMURF1 depletion resulted in a modest increase in FTH1 levels, an effect that was more pronounced following erastin treatment. Similarly, SMURF1 knockdown extended the half-life of FTH1, with a more significant impact observed after erastin exposure (Figure 5C).

Subsequently, we investigated the impact of SMURF1 overexpression on FTH1 expression levels in myoblasts. In comparison to the control group, the overexpression of SMURF1 led to a reduction in FTH1 protein levels while having no discernible effect on FTH1 mRNA levels (Figure 6A,B). This effect was observed irrespective of the presence or absence of Ferrostatin-1 (Fer-1). Next, we aimed to ascertain if SMURF1 plays a role in regulating the ubiquitination and subsequent degradation of FTH1. When compared to the control shRNA, the knockdown of SMURF1 notably reduced FTH1 ubiquitination while concurrently elevating FTH1 protein levels (as illustrated in Figure 6C). Conversely, the overexpression of SMURF1 led to a significant increase in FTH1 ubiquitination and a corresponding decrease in FTH1 protein levels (as depicted in Figure 6D). These findings suggest that SMURF1 regulates FTH1 protein levels through its ubiquitin ligase activity.

### 2.6. SMURF1 Regulates Ferroptosis via FTH1 to Affect Myoblast-to-Myotube Differentiation

We then investigated whether SMURF1 could selectively target FTH1, thereby inhibiting ferroptosis and promoting skeletal muscle development. We co-transfected an overexpression vector of FTH1 and 2 shRNA vector targeting FTH1 and SMURF1. The experimental group details are presented in Figure 7A. We observed that the knockdown of SMURF1 markedly suppressed ferroptosis in mouse myoblasts. Notably, subsequent transfection with FTH1 shRNA further augmented ferroptosis, whereas the overexpression of FTH1 attenuated this process (Figure 7B–F). Immunofluorescence staining for MyHC demonstrated that SMURF1 knockdown enhanced myotube differentiation in mouse myoblasts, but this effect was significantly impaired by co-transfection with FTH1 shRNA and promoted by FTH1 overexpression via plasmid transfection (Figure 7G). The qPCR analysis revealed that the SMURF1 knockout upregulated MyoG and MyoD mRNA expression in mouse myoblasts, an effect that was significantly suppressed by co-transfection with FTH1 shRNA but restored by FTH1 overexpression via plasmid transfection (Figure 7H,I). Our findings suggest that SMURF1 plays a crucial role in regulating myogenesis through its mediation of FTH1 expression.

## 3. Discussion

Skeletal muscle plays a pivotal role in locomotion, metabolism, and overall physiological homeostasis. Investigating the molecular mechanisms of skeletal muscle development is essential for unraveling the complex processes that underpin muscle formation and function [28,29]. This research not only deepens our understanding of developmental biology but also holds promise for the development of novel therapies for muscle-related diseases, ultimately contributing to enhanced human health and performance.

Iron plays a fundamental role in regulating muscle differentiation and maintaining muscle phenotype, as evidenced by a series of recent studies. Iron is an essential component of numerous enzymes involved in cellular processes, including those related to myogenesis. It was demonstrated that iron deficiency can significantly impair muscle development and differentiation, leading to a decrease in muscle mass and altered muscle fiber composition [30,31]. Specifically, Che et al. (2023) reported that iron-deficient mice exhibited delayed muscle regeneration and the reduced expression of myogenic regulatory factors, highlighting the importance of iron in the myogenic program [32]. Furthermore, iron homeostasis is crucial for maintaining the functional properties of mature muscle fibers. Excessive iron accumulation, on the other hand, has been linked to muscle dysfunction and the development of muscle-related disorders, possibly through the generation of reactive oxygen species and subsequent oxidative stress [33]. Fefelova et al. (2023) found that iron overload in muscle cells led to impaired mitochondrial function and decreased muscle contractility, underscoring the need for the tight regulation of iron levels in muscle tissue [33]. These findings emphasize the critical role of iron in muscle differentiation and phenotype maintenance. At the molecular level, iron homeostasis in skeletal muscle is regulated by a complex network of proteins [34]. In addition to transferrin receptor protein 1 and solute carrier family 40 member 1, other key regulators include divalent metal transporter 1, which facilitates iron influx into the cell, and hepcidin, which downregulates iron export by binding to FPN1 [35]. Recent research has also highlighted the role of iron-regulatory proteins like IRP1 and IRP2, which bind to iron-responsive elements in mRNA and regulate the translation of iron-related proteins [36]. Moreover, iron overload in skeletal muscle can lead to oxidative stress and muscle damage [37]. The accumulation of iron can catalyze the formation of reactive oxygen species, which damage cellular components and impair muscle function. Therefore, maintaining iron homeostasis in muscle is crucial for preserving muscle health and function [38]. Therefore, iron is essential for skeletal muscle function, and its metabolism is tightly regulated at the molecular level. Disruptions in iron homeostasis can significantly impact muscle performance, making the study of iron regulation in skeletal muscle a crucial area of research. FTH1 is highly expressed in the liver, spleen, and bone marrow—the primary tissues responsible for iron storage and regulation. Within these tissues, FTH1 serves as a pivotal component of ferritin, the chief intracellular protein for iron storage [27]. *FTH1* facilitates the oxidation of ferrous iron (Fe^2+^) to ferric iron (Fe^3+^) and its subsequent incorporation into the ferritin core, thereby playing a crucial role in maintaining iron homeostasis. When cellular iron levels are elevated, ferritin synthesis increases, effectively sequestering excess iron. Conversely, during periods of iron deficiency, ferritin undergoes degradation, releasing stored iron for cellular utilization. In the liver, FTH1’s function is particularly critical as it protects cells from the toxic effects of free iron by encapsulating it within the ferritin complex [39]. This is especially important given the liver’s susceptibility to iron overload conditions. In the brain, FTH1 plays a vital role in neurotransmission, myelination, and energy metabolism. The accumulation of iron in the brain, as observed in neurodegenerative diseases such as Parkinson’s and Alzheimer’s, has been linked to dysfunction in *FTH1* [11,40]. This underscores the significance of *FTH1* in maintaining iron homeostasis within the central nervous system. In muscle and heart, *FTH1* contributes to the regulation of iron levels, which are essential for muscle contraction and energy metabolism [24]. Iron deficiency can impair muscle function and lead to anemia, further highlighting the importance of FTH1 in maintaining iron homeostasis in this tissue [41]. Furthermore, FTH1 has been implicated in the immune response, particularly in macrophages [42]. Macrophages upregulate FTH1 expression during infection to sequester iron, thereby depriving pathogens of this essential nutrient and enhancing microbial killing [43]. Collectively, these observations demonstrate that FTH1 plays a multifaceted role in iron metabolism and homeostasis across various tissues. Alterations in FTH1 expression are closely associated with the pathogenesis of several diseases. Therefore, a deeper understanding of the molecular regulatory mechanisms governing *FTH1* is imperative for the development of novel therapeutic strategies to combat these conditions. In our study, we observed a notable activation of ferroptosis in myoblasts derived from mice with a skeletal muscle-specific knockout of *FTH1*. This activation resulted in a significant inhibition of GSH metabolism, concurrently promoting elevated levels of lipid peroxidation and iron accumulation. Consequently, these alterations led to the suppression of skeletal muscle atrophy and the manifestation of muscle weakness. These findings underscore the critical role of *FTH1* in maintaining iron homeostasis and preventing ferroptosis in skeletal muscle cells, thereby preserving muscle health and function.

*SMURF1* is a member of the HECT-type E3 ubiquitin ligase family, functions as a key regulator of multiple cellular signaling pathways [44]. It primarily targets proteins for ubiquitination and subsequent degradation, thereby modulating their stability and activity. *SMURF1* plays crucial roles in bone homeostasis, cell polarity, and cancer progression [45]. One of the primary regulatory mechanisms of SMURF1 involves its interaction with specific proteins, such as mothers against decapentaplegic homolog 1 (Smad1), mothers against decapentaplegic homolog 5 (Smad5), mitogen-activated protein kinase kinase kinase 2 (MEKK2), and transforming protein RhoA (RhoA), which leads to their ubiquitination and degradation [46,47]. For instance, SMURF1 promotes bone formation by inhibiting the BMP signaling pathway through ubiquitination and degradation of Smad1 and Smad5 [48]. Additionally, SMURF1 regulates cell polarity and motility by controlling the level of RhoA [49]. Considering the crucial roles of iron homeostasis and lipid peroxidation in ferroptosis, a unique type of cell death triggered by iron-mediated lipid peroxidation, it is conceivable that SMURF1 could be involved in modulating this process. Ferroptosis is marked by an increase in lipid peroxidation byproducts and the impaired function of glutathione peroxidase 4 [50]. SMURF1 could potentially influence ferroptosis by modulating the stability and activity of proteins involved in iron metabolism or the antioxidant defense system [51]. Gu et al. found that TUG1 promotes the ubiquitination and breakdown of growth differentiation factor 15 through a direct interaction with SMURF1, leading to the inhibition of nuclear factor erythroid 2-related factor 2 and subsequently suppressing ferroptosis [52]. However, direct evidence linking SMURF1 to ferroptosis regulation remains scarce. Future studies exploring the interaction between SMURF1 and proteins involved in iron metabolism or the antioxidant defense system could provide valuable insights into the potential role of SMURF1 in ferroptosis. In this study, we have identified SMURF1 as a bona fide E3 ubiquitin ligase for FTH1, which facilitates the degradation of FTH1 protein through the ubiquitination pathway. This process plays a crucial role in regulating lipid peroxidation levels and iron homeostasis within skeletal muscle, thereby exerting significant control over skeletal muscle health. FTH1, as an integral component of ferritin, engages in a crucial interaction with NCOA4, playing a pivotal role in the process of ferritin phagocytosis [11,14]. NCOA4 acts as a selective autophagy cargo receptor, binding specifically to FTH1 within the autophagosome and facilitating its delivery to the lysosome, where iron is subsequently released [53]. This intricate process is indispensable for maintaining the delicate iron balance within cells. SMURF-1 is ubiquitously expressed in the cytoplasm, where it functions as an E3 ubiquitin ligase with the capability to specifically recognize and bind to target proteins [34]. It catalyzes the covalent attachment of ubiquitin molecules to these target proteins, forming ubiquitination modifications that serve as tags for subsequent degradation [35]. These ubiquitinated proteins are then recognized and efficiently degraded by the proteasome. Through this intricate mechanism, SMURF-1 exerts precise control over the degradation rates of specific proteins within the cytoplasm, thereby influencing a myriad of cellular physiological functions, including cell proliferation, differentiation, and migration [54]. Notably, by interacting with and targeting the FTH1 protein for degradation, SMURF-1 reduces the expression level of FTH1. This, in turn, disrupts the delicate balance of iron storage, iron release, anti-oxidative stress response, and iron metabolism regulation pathways, ultimately accelerating the onset of cellular death. Consequently, SMURF1 may exert an influence by promoting the degradation of FTH1 through the deubiquitination pathway. This, in turn, diminishes the expression levels of FTH1, thereby disrupting the homeostasis of ferritinophagy in skeletal muscle. Such disruptions can lead to the impairment of skeletal muscle health and function, underscoring the importance of tightly regulated ferritinophagy in preserving muscular integrity. Such research may also provide insights into diseases linked to iron dysregulation, holding significant implications for skeletal muscle health, athletic performance, and studies on aging and disease.

## 4. Materials and Methods

### 4.1. Animals

This animal study was authorized by the Sichuan Academy of Animal Science, Chengdu, China, adhering to ethical guidelines. We used eight-week-old C57B/6J mice, with each group containing three males and three females. Beijing Biocytogen Co. Ltd. developed and electroporated a targeting vector containing loxP sites and an Frt-flanked neomycin cassette into C57BL/6J embryonic stem cells to produce the FTH^flox/flox^ mouse model. In order to specifically delete the FTH gene in skeletal muscle for functional study, FTH1flox/flox mice were subsequently crossed with Jackson Laboratory Mrf5-Cre mice to create skeletal muscle-conditional FTH knockout mice.

### 4.2. In Vitro Cell Culture and Differentiation

A conventional procedure was used to separate skeletal myoblasts from the hind limbs of 1-week-old C57B/6J mice [55]. In Ham’s F-10 medium, which was supplemented with 10% horse serum and antibiotics, muscles were treated with 700–800 U/mL collagenase II at 37 °C. Following washing, cells underwent centrifugation, collagenase inactivation treatment, and subsequent incubation. After impurities were eliminated by filtration and pre-plating, myoblasts were purified by antibody labeling and sorting. After that, cells were cultivated at 37 °C with 5% CO_2_ in DMEM containing 10% fetal bovine serum, glutamine, and antibiotics. Myoblasts were sown in 24-well plates for differentiation, and, when they reached 80% confluence, the media was changed to DMEM with 5% horse serum.

### 4.3. RNA Extraction and Real-Time PCR Analysis

Following the manufacturer’s instructions, TRIzol Reagent (TaKaRa) was used to isolate total RNA from tissues or cells [56]. The PrimeScript RT reagent Kit with gDNA Eraser (Takara) was then used to reverse-transcribe 1 μg of RNA into cDNA. Following the manufacturer’s instructions, 1 μL of a 1:5 cDNA dilution was used with SYBR Premix Ex Taq II (TliRNaseH Plus) (Takara) for PCR amplification. A 30 s initial denaturation at 95 °C was followed by 40 cycles of 95 °C for 5 s and 60 °C for 30 s as part of the thermal cycling conditions. Appendix A contains information on real-time PCR primers. According to MIQE guidelines, the comparative threshold cycle^(ΔΔCT)^ approach was used to calculate relative expression [57].

### 4.4. Cell Transfection

pcDNA3.1 vectors carrying mouse FTH1 or SMURF1 fused to Flag tags (OriGene, Rockville, MD, USA) for overexpression, and shRNAs targeting these genes (Sigma-Aldrich, St. Louis, MO, USA) for silencing were used to stably transfect C2C12 cells. As directed by the manufacturer, stable clones were produced using the FuGENE reagent (Promega, Milan, Italy). In short, 10,000 cells per well were planted onto six-well plates, and, the next day, the cells were transfected. After five minutes of mixing FuGENE with serum-free DMEM, shRNA, overexpression, or scrambled control plasmids were added. Following a 30 min incubation period, cells were cultivated for 48 h prior to selection using either G418 (for overexpressing clones) or 3 μg/mL of puromycin (for silent clones). After pooling resistant cells and confirming their changed expression of FTH1 or SMURF1 using Western blot and qRT-PCR, they were employed.

### 4.5. Protein Extraction and Western Blot Analysis

Cells were collected via centrifugation at 700× *g* for 10 min at 4 °C and then lysed in 50 μL of a buffer solution (Sigma-Aldrich). After incubating on ice for 30 min, the lysates were cleared through centrifugation at 4 °C for 10 min, retaining the supernatants. Protein levels were determined using a Bradford assay kit (Sigma-Aldrich). Equal protein quantities were separated by SDS-PAGE, with Kaleidoscope prestained standards (Bio-Rad, Hercules, CA, USA) serving as molecular weight markers. The proteins were subsequently transferred onto PVDF membranes via electroblotting. Following blocking with 5% milk, the membranes were incubated with primary antibodies overnight. Detection of the target protein was achieved using the appropriate secondary antibody conjugated to horseradish peroxidase for 1 h, followed by visualization through enhanced chemiluminescence. Band densitometry, normalized to α-tubulin, was carried out using ImageJ software 2.16.0 (developed by the National Institutes of Health, Bethesda, MD, USA). The primary antibodies employed included the following: rabbit anti-SMURF1 (Sigma-Aldrich), mouse anti-MHC (Novus Biologicals, St. Louis, MO, USA), rabbit anti-FTH1 (MBL, Woburn, MA, USA), rabbit anti-ACSL4 (Santa Cruz Biotechnology, Santa Cruz, CA, USA), rabbit anti-COX2 (Santa Cruz Biotechnology, Santa Cruz, CA, USA), mouse anti-ubiquitin (Abcam, London, Cambridge, UK), mouse anti-Flag (Abcam), and rabbit anti-GAPDH (Sigma-Aldrich). The secondary antibodies used were the following: mouse anti-rabbit HRP and goat anti-mouse HRP from ABclonal Technology Inc. (Beijing, China).

### 4.6. Immunofluorescence Staining

Cells were preserved using a 4% paraformaldehyde solution for a duration of 20 min and then subjected to permeabilization with 0.25% Triton X-100 for 10 min at ambient temperature. Next, the cells underwent a blocking step with 1% BSA for half an hour at room temperature. This was followed by an overnight incubation at 4 °C, with gentle shaking, using a primary antibody specific to MyHC (MF20, diluted to a ratio of 1:400, sourced from the Developmental Studies Hybridoma Bank, University of Iowa). Post-incubation, the cells were rinsed thrice with PBS. Subsequently, they were exposed to a fluorescein isothiocyanate-conjugated secondary antibody (diluted 1:100, Cell Signaling Technology, Danvers, MA, USA) for an hour at room temperature and then washed again three times with PBS. For nuclear visualization, DAPI staining was applied. The immunofluorescence imaging was captured utilizing a Leica Q500MC imaging system equipped for fluorescence microscopy.

### 4.7. Immunoprecipitation Assay

Protein extraction was carried out following established protocols. The protein content was quantified using a Bradford assay kit, and the lysates were standardized to a concentration of 1 mg/mL using a lysis buffer containing phosphatase inhibitors. For preclearing, 5 μg of non-specific IgG (Sigma, St. Louis, MO, USA) and 20 μL of protein A/G plus-agarose beads (Sigma) were added to the lysates and incubated at 4 °C for 1 h with gentle rotation. Following centrifugation at 500× *g* for 5 min at 4 °C, 10 μg of either the SMURF1 or FTH1 antibody, or the appropriate IgG control, was introduced to the precleared lysates and incubated on ice for 5 h. Thereafter, 30 μL of protein A/G plus-agarose beads were added, and the samples were rotated overnight at 4 °C. The bound complexes were then isolated by centrifugation, and the resulting pellets were washed four times with PBS containing phosphatase inhibitors (PBS-PIs) before being resuspended in 20 μL of the sample loading buffer. The samples were subsequently separated by electrophoresis on a 12% SDS-PAGE gel and subjected to immunoblotting with the specified antibodies, enabling the specific detection and analysis of the proteins of interest.

### 4.8. Assessment of Lipid Peroxidation, GSH Levels, and Iron Content

For the quantification of lipid peroxidation, GSH levels, and iron content, we followed manufacturer protocols using specific kits. Lipid peroxidation products, MDA and 4-HNE, were measured in cell lysates with kits from Abcam (ab118970 and ab238538). GSH levels were assessed using a Sigma-Aldrich GSH assay kit (CS0260). Cell lysates were prepared with a detergent-containing buffer and protease inhibitors, then processed according to the kit instructions. Iron content was determined with a Sigma iron assay kit (Sigma-Aldrich). Cell lysates were similarly prepared and processed as for the GSH measurement. All measurements included appropriate controls and were performed in triplicate for reproducibility.

### 4.9. Statistical Analyses

In our analysis, we applied the two-tailed Student’s *t*-test to evaluate the statistical significance between pairwise comparisons. For multi-group comparisons, we conducted ANOVA followed by Tukey’s HSD post hoc test. All statistical analyses were performed using JMP 17 software (SAS, Cary, NC, USA). Statistical significance was determined at a *p* value threshold of <0.05. Data are reported as means ± standard error of the mean (S.E.M.).

## 5. Conclusions

In this study, we found a close relationship between ferroptosis and skeletal muscle development and found that SMURF1 affects the stability of FTH1 protein through its interaction with FTH1, thus affecting the iron homeostasis of skeletal muscle, which has an important role in maintaining the function and health of skeletal muscle. A deeper understanding of the key genes and targets involved in ferroptosis not only offers novel therapeutic and preventive approaches for skeletal muscle diseases but also provides theoretical foundations for other diseases, such as cancer and neurodegenerative disorders.

## Figures and Tables

**Figure 1 ijms-26-01390-f001:**
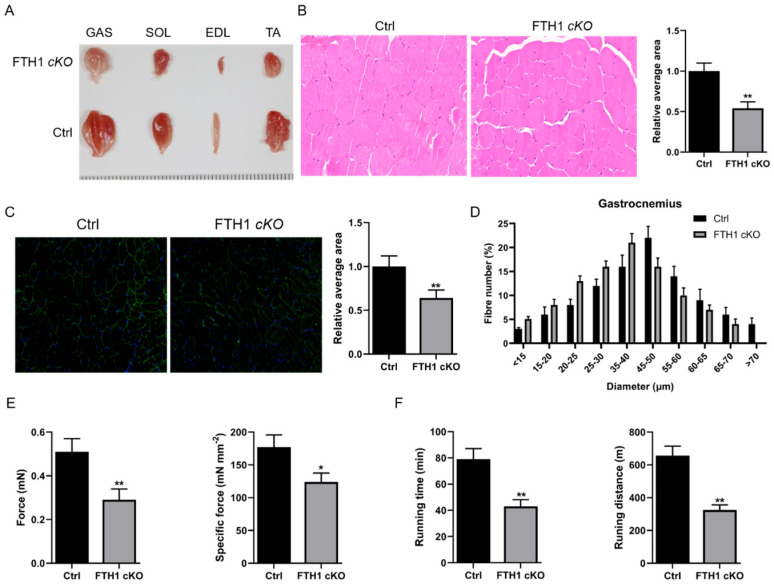
Morphological and functional alterations in FTH1 knockout mice: (**A**) Representative images of gastrocnemius (GAS), soleus (SOL), extensor digitorum longus (EDL), and tibialis anterior (TA) muscles from control and *FTH1* conditional knockout (cKO) mice. (**B**) Histological examination of GAS muscle sections stained with hematoxylin and eosin. The mean cross-sectional area of muscle fibers is presented on the right. (**C**) Immunohistochemical staining of GAS muscle fibers from mice using an anti-dystrophin antibody. The relative mean area is displayed on the right. The green is the dystrophin protein, and the blue is the nucleus. (**D**) Size distribution and the average diameter of muscle fibers in the gastrocnemius muscle of *FTH1* cKO mice compared to control mice. (**E**) Force measurements obtained during the tetanic contraction of GAS muscles from control and *FTH1* cKO mice. (**F**) Assessment of muscle endurance through forced treadmill running to exhaustion. Data are reported as means ± standard error of the mean (s.e.m.). * indicates *p* < 0.05; ** indicates *p* < 0.01.

**Figure 2 ijms-26-01390-f002:**
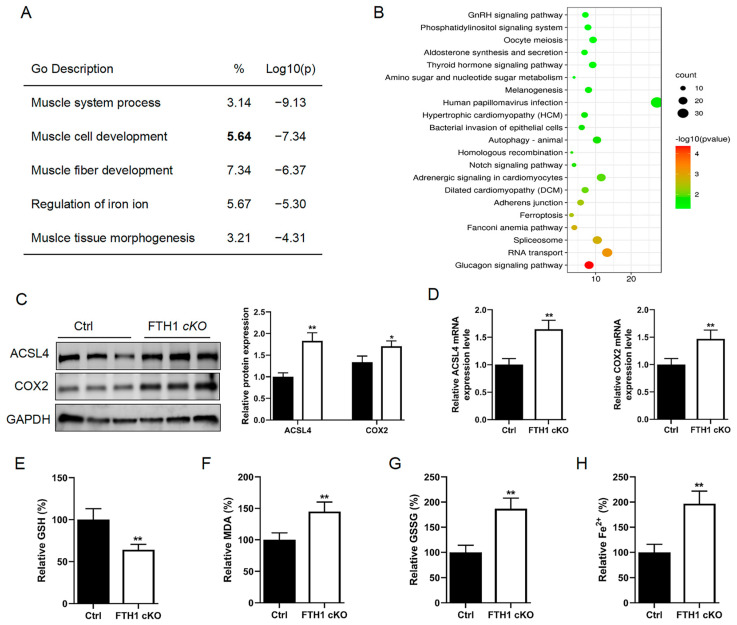
Impact of FTH1 on ferroptosis in mouse myoblast cells: (**A**) Gene Ontology (GO) enrichment analysis conducted to identify significantly differentially expressed genes between *FTH1* cKO myoblasts and their control counterparts. (**B**) Pathway enrichment analysis performed to elucidate the pathways associated with the significantly differentially expressed genes observed between *FTH1* cKO myoblasts and controls. (**C**) Western blot analysis employed to assess the expression levels of ferroptosis-related proteins ACSL4 and COX2 in FTH1 cKO myoblasts compared to controls. (**D**) qPCR utilized to measure the mRNA expression levels of ACSL4 and COX2 in *FTH1* cKO myoblasts relative to control cells. (**E**–**H**) The concentrations of GSH, MDA, and Fe^2+^ quantified and compared between *FTH1* cKO myoblasts and control myoblasts to determine their relative levels. Data are reported as means ± standard error of the mean (s.e.m.). * indicates *p* < 0.05; ** indicates *p* < 0.01.

**Figure 3 ijms-26-01390-f003:**
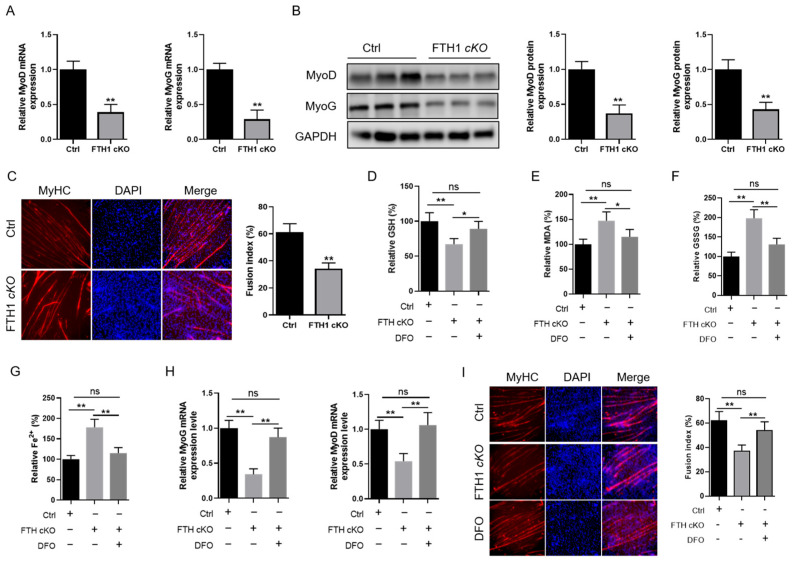
FTH1 regulates myoblast differentiation into myotubes via the ferroptosis pathway: (**A**) qPCR analysis of MyoG and MyoD mRNA expression in FTH1 cKO myoblasts relative to control cells. (**B**) Western blot assessment of MyoD protein levels in FTH1 cKO myoblasts compared to controls. (**C**) MyHC immunofluorescence staining to determine the myotube fusion index in FTH1 cKO myoblasts versus controls. Red: MyHC, Green, nucleus. (**D**–**G**) Quantification of GSH, MDA, and Fe^2+^ concentrations in FTH1 cKO myoblasts and control myoblasts, with and without DFO treatment. (**H**) qPCR analysis of MyoG and MyoD mRNA expression in FTH1 cKO myoblasts relative to controls, with and without DFO treatment. (**I**) MyHC immunofluorescence staining to assess the myotube fusion index in FTH1 cKO myoblasts compared to controls, with and without DFO treatment. Red: MyHC, Green, nucleus. Data are reported as means ± standard error of the mean (s.e.m.). * indicates *p* < 0.05; ** indicates *p* < 0.01, ns: not significant.

**Figure 4 ijms-26-01390-f004:**
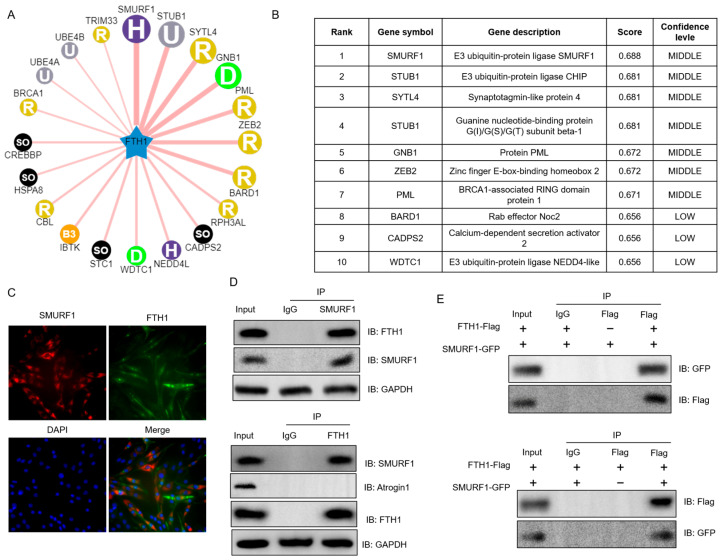
SMURF1 identified as the E3 ligase for FTH1: (**A**,**B**) Prediction of SMURF1 as the specific E3 ligase for FTH1 using the UbiBrowser database. (**C**) Confocal microscopy images showing the colocalization of SMURF1 (red) and FTH1 (green). (**D**) Reciprocal co-immunoprecipitation analysis revealing the interaction between SMURF1 and FTH1 in primary myoblasts cultured for 4 days in a differentiation medium. Atrogin1 used as native control, GAPDH used as loading control. (**E**) Validation of the interaction by reciprocal co-immunoprecipitation between GFP-tagged SMURF1 and Flag-tagged FTH1 in HEK293T cells. Data are reported as means ± standard error of the mean (s.e.m.).

**Figure 5 ijms-26-01390-f005:**
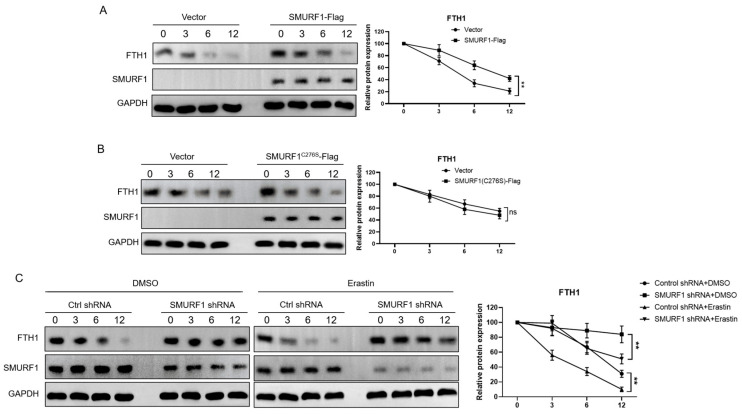
SMURF1 promotes the degradation of FTH1: (**A**,**B**) The wild type SMURF1 (**A**) decreases the FTH1 protein level, but the C276S mutant (**C**) cannot downregulate FTH1. Cells expressing Flag-SMURF1 or Flag-SMURF1C276S treated with cycloheximide (CHX, 200 μg/mL). (**C**) SMURF1 is critical to the stability of FTH1 before or after erastin treatment. Cells expressing indicated shRNA constructs were treated with cycloheximide (CHX, 200 μg/mL) and erastin (0–12 h, 5 μM). Data are reported as means ± standard error of the mean (s.e.m.). ** indicates *p* < 0.01. ns: not significant.

**Figure 6 ijms-26-01390-f006:**
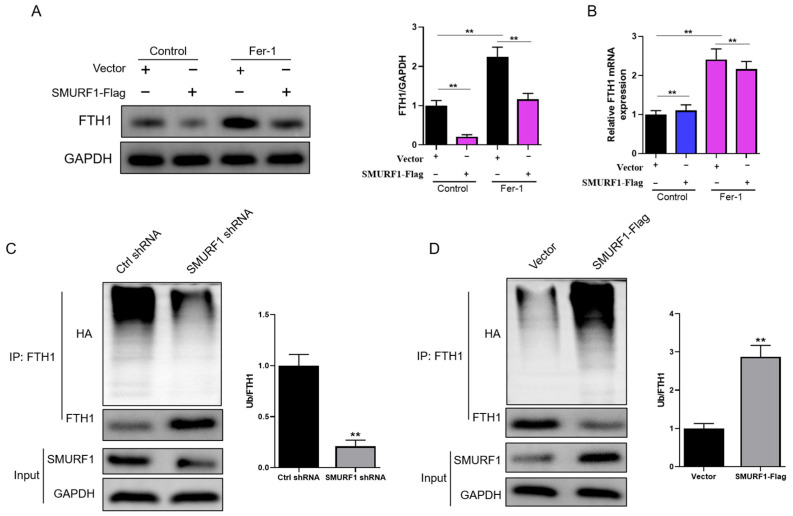
SMURF1 promotes FTH1 ubiquitination and degradation: (**A**) Representative Western blot analysis and quantification of FTH1 protein levels in cells treated with Fer-1 (5 µM) for 12 h. (**B**) qPCR analysis of FTH1 mRNA expression in cells treated with Fer-1 (5 µM) for 12 h. (**C**) The immunoprecipitation (IP) of lysates from HEK 293T cells transfected with either control (Ctrl) or SMURF1-shRNA and pretreated with MG132 prior to collection, followed by detection with the indicated antibodies. (**D**) The IP of lysates from HEK 293T cells transfected with either Vector or SMURF1 overexpression constructs and pretreated with MG132 prior to collection, followed by detection with the indicated antibodies. Data are reported as means ± standard error of the mean (s.e.m.). ** indicates *p* < 0.01.

**Figure 7 ijms-26-01390-f007:**
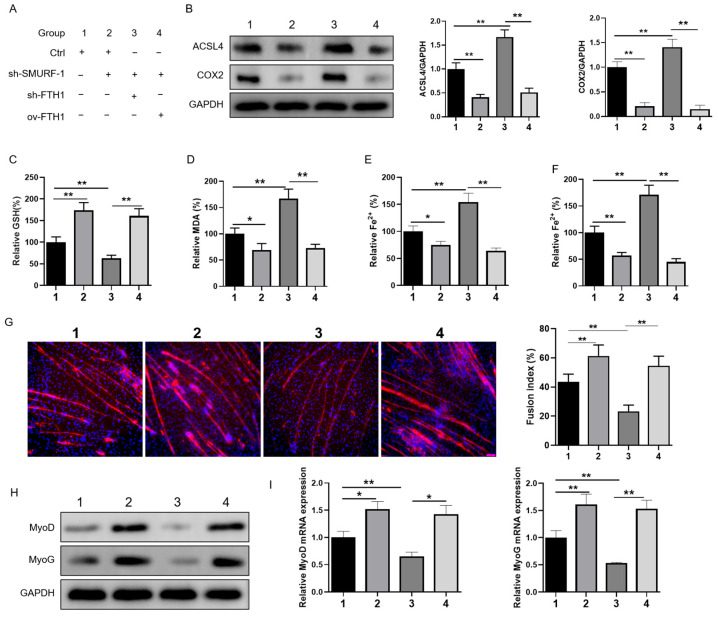
SMURF1 targets FTH1 to facilitate ferroptosis and impede skeletal muscle development: (**A**) Overview of the experimental groups and treatments employed in this study phase. (**B**) Western blot analysis depicting the protein levels of ACSL4 and COX2 in the specified cell treatments. (**C**–**F**) Quantification of GSH, MDA, GSSG, and Fe^2+^ concentrations in the indicated treated cells. (**G**) Immunofluorescence staining for MyHC in the specified cell treatments. Red: MyHC, Green, nucleus. (**H**,**I**) Western blot analysis of MyoG and MyoD protein expression in the indicated treated cells. (**I**) qRT-PCR analysis of MyoG and MyoD mRNA expression in the indicated treated cells. * indicates *p* < 0.05; ** indicates *p* < 0.01.

## Data Availability

The raw data supporting the conclusions of this article will be made available by the authors on request.

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
