# Peer review of "SMURF1-Induced Ubiquitination of FTH1 Disrupts Iron Homeostasis and Suppresses Myogenesis"

_ijms, 2025, doi:10.3390/ijms26031390_

Round 1
Reviewer 1 Report
Comments and Suggestions for Authors
In the article entitled “SMURF1-Induced Ubiquitination of FTH1 Disrupts Iron Homeostasis and Suppresses Myogenesis” by Xia Xiong et al., the authors evaluated the effects of modulating FTH1 activity in the contest of muscle development and regeneration. The work is interesting, and the experiments are well performed, but the paper needs to be rewritten because it is difficult to understand.
The abstract and introduction are confusing. The main focus of the work and the role of FTH1 in muscle development and regeneration are poorly delineated, as is the regulation of the protein also by ubiquitination.
Major
Demonstration that the animals are FTH1 cKO is lacking
Some of the information reported in the discussion should be included in the introduction.
Although the experiments appear interesting and well conducted, the work is presented unclearly. The purpose of the work and the meaning and role of the factors that the authors determine are not exhaustively indicated. Much information is missing on the role of iron regulation on muscle development and homeostasis. In particular, there is a lack of information relating to the role of the FTH1 protein in these processes, just as there is a lack of information on the regulation of the FTH1 protein itself.
Minor
In Figure 2, panel c does not have what the columns refer to.
The authors must define the meaning of the acronyms they use, such as TfR1, FPN1 DMT1 IRP2, BMP signaling Smad 1, Smad5….
Author Response
Dear Editors and Reviewers,
Thank you for your letter and for the reviewers’ comments concerning our manuscript entitled “SMURF1-Induced Ubiquitination of FTH1 Disrupts Iron Homeostasis and Suppresses Myogenesis” (No. ijms-3389752). These comments are all valuable and very helpful for revising and improving our paper, as well as having significant guiding significance for our research. We have carefully studied the comments and made the necessary corrections accordingly. The modifications and revisions are marked in yellow in the manuscript, and below are the responses to your comments and suggestions for your review.
Reviewer 1:
Major
Comments 1. Demonstration that the animals are FTH1 cKO is lacking.
Response: Thank you for your suggestion. We used qPCR to analyze FHT1 mRNA expression levels in tissues of FHT1 cKO mice and control mice. FTH1 was found to be virtually unexpressed in the skeletal muscle of FTH-knocked out mice and unchanged in other tissues.
Comments 2. Some of the information reported in the discussion should be included in the introduction.
Thank you for your suggestion. We have included some of the reporting information from the discussion in the introduction to fill in what is missing from the introduction in line 64-70.
Comments 3: Although the experiments appear interesting and well conducted, the work is presented unclearly. The purpose of the work and the meaning and role of the factors that the authors determine are not exhaustively indicated. Much information is missing on the role of iron regulation on muscle development and homeostasis. In particular, there is a lack of information relating to the role of the FTH1 protein in these processes, just as there is a lack of information on the regulation of the FTH1 protein itself.
Response: Thank you for your suggestion. We have added some information about the role of iron in regulating muscle development and homeostasis. In addition, the information on FTH1 regulation of skeletal muscle development was supplemented in line 77-83.
Minor
Comments 4: In Figure 2, panel c does not have what the columns refer to.
Response: Thank you for your suggestion. We have revised them in line 125-128.
Comments 5: The authors must define the meaning of the acronyms they use, such as TfR1, FPN1 DMT1 IRP2, BMP signaling Smad 1, Smad5….
Response: Thank you for your suggestion. We have defined the meaning of the acronyms in line 16, 25, 37, 38, 39, 127, 277, 279, 302 and 304.
Reviewer 2 Report
Comments and Suggestions for Authors
The work shows that conditional muscle FTH-KO mice have morphological and functional alterations in the muscles. Then the authors analyze the myoblasts of the mice, to find that they have signs of ferroptosis that can be alleviated by treatment with iron chelator. Since the ferroptosis is linked to FTH deprivation, the authors look for actors that may reduce FTH, and in silico research suggested that the ubiquitin ligase SMURF1 may bind FTH. They confirmed the binding and that SMURF1 modulated ferritin protein level. The work is novel, but I have some concerns.
- MCK is expressed in skeletal muscles and the heart, thus the reduced force and endurance of the KO mice may be caused by heart insufficiency. This should be discussed.
- Evidence of the absence of FTH in the muscles and the myoblasts should be presented. In fact, ferritin expression is shown in Fig. 5, which refers to the control mice's myoblasts, I guess.
- Ferritinophagy is the best-characterized mechanism of ferritin degradation, which involves FTH binding NCOA4. The authors should comment on why they did not consider ferritinophagy but looked for alternative ubiquitin-proteosome pathways of ferritin degradation. Do they have evidence for cell-specific differences in ferritin degradation?
- The discussion should describe and comment on the proposed mechanism that involves SMURF1, FTH, iron and ferroptosis.
Author Response
Dear Editors and Reviewers,
Thank you for your letter and for the reviewers’ comments concerning our manuscript entitled “SMURF1-Induced Ubiquitination of FTH1 Disrupts Iron Homeostasis and Suppresses Myogenesis” (No. ijms-3389752). These comments are all valuable and very helpful for revising and improving our paper, as well as having significant guiding significance for our research. We have carefully studied the comments and made the necessary corrections accordingly. The modifications and revisions are marked in yellow in the manuscript, and below are the responses to your comments and suggestions for your review.
Comments 1. MCK is expressed in skeletal muscles and the heart, thus the reduced force and endurance of the KO mice may be caused by heart insufficiency. This should be discussed.
Response: We offer our sincere apologies for the error and express gratitude to the reviewer for bringing it to our attention. We used the Myf5-Cre not MCK-Cre mice to generate the Nestin mice with a conditional knockout of skeletal muscle. This has been revised in Line 346.
Comments 2. Evidence of the absence of FTH in the muscles and the myoblasts should be presented. In fact, ferritin expression is shown in Fig. 5, which refers to the control mice's myoblasts, I guess.
Response: Thank you for your comments. To facilitate easier comparison and manipulation of cells, we utilized a myoblast cell line derived from C2C12 mice and transfected them with SMURF1 interference vectors (SMURF1 shRNA) and overexpression vectors (SMURF1-Flag), aiming to investigate their impacts on FTH1 protein expression.
Comments 3. Ferritinophagy is the best-characterized mechanism of ferritin degradation, which involves FTH binding NCOA4. The authors should comment on why they did not consider ferritinophagy but looked for alternative ubiquitin-proteosome pathways of ferritin degradation. Do they have evidence for cell-specific differences in ferritin degradation?
Response: Thank you for your comments. Thank you for your insightful comments. As you mentioned, FTH1, serving as a crucial subunit of ferritin, works in close collaboration with NCOA4 to modulate iron metabolism and maintain cellular iron balance during ferritinophagy. This topic is indeed fascinating. Currently, we are delving deeper into the role of SMURF1 in regulating the ferritinophagy pathway by modulating FTH1 in skeletal muscle. Our aim is to further elucidate the molecular mechanism by which FTH1 regulates ferritinophagy in this specific tissue context.
Comments 4. The discussion should describe and comment on the proposed mechanism that involves SMURF1, FTH, iron and ferroptosis.
Response: Thank you for your suggestion. We have discussed it in line 325-335.
Round 2
Reviewer 1 Report
Comments and Suggestions for Authors
The authors have responded satisfactorily to all the comments I have made. I therefore recommend publishing the article.
Author Response
Dear reviewer, thank you for taking the time to review my manuscript and provide constructive comments. Your feedback has been invaluable in improving the quality of the work.
Reviewer 2 Report
Comments and Suggestions for Authors
The answers to the points I raised are not satisfactory.
- Figure 1 should show evidence that muscles of FTH1 KO mice do not express FTH1. A western blotting would be adequate.
More detailed comments should be presented on the proposed proteasomal degradation pathway in which SMURF-1 is involved, particularly how the ferritin iron core could be handled after protein degradation in the cytosol.
- In the results some details on the characteristics of the Myf5-Cre mice.
Author Response
Comments 1: Figure 1 should show evidence that muscles of FTH1 KO mice do not express FTH1. A western blotting would be adequate.
Response: Thank you for your suggestion. We have presented evidence demonstrating the absence of FTH1 expression in the muscle of FTH1 KO mice, as illustrated by the western blot in Fig. S1C.
Comments 2: More detailed comments should be presented on the proposed proteasomal degradation pathway in which SMURF-1 is involved, particularly how the ferritin iron core could be handled after protein degradation in the cytosol.
Response: Thank you for your suggestion. We have provided a detailed elucidation of the function of SMURF1 within the proteasome degradation pathway, and delve into the mechanisms by which SMURF1 modulates the stability of the FTH1 protein, thereby influencing cell fate in line 3321-342.
Comments 3: In the results some details on the characteristics of the Myf5-Cre mice.
Response: Thank you for your suggestion. We have revised the details regarding the characteristics of the Myf5-Cre mice in lines 95-97.
Round 3
Reviewer 2 Report
Comments and Suggestions for Authors
The answers to the points I raised are acceptable
Author Response
Thank you very much for your recognition and dedication to our research.